# Non-Ising and chiral ferroelectric domain walls revealed by nonlinear optical microscopy

Salia Cherifi-Hertel[1], Hervé Bulou[1], Riccardo Hertel[1], Grégory Taupier[1], Kokou Dodzi (Honorat) Dorkenoo[1], Christian Andreas[1], Jill Guyonnet[2], Iaroslav Gaponenko[2], Katia Gallo[3] & Patrycja Paruch[2]

The properties of ferroelectric domain walls can significantly differ from those of their parent material. Elucidating their internal structure is essential for the design of advanced devices exploiting nanoscale ferroicity and such localized functional properties. Here, we probe the internal structure of 180° ferroelectric domain walls in lead zirconate titanate (PZT) thin films and lithium tantalate bulk crystals by means of second-harmonic generation microscopy. In both systems, we detect a pronounced second-harmonic signal at the walls. Local polarimetry analysis of this signal combined with numerical modelling reveals the existence of a planar polarization within the walls, with Néel and Bloch-like configurations in PZT and lithium tantalate, respectively. Moreover, we find domain wall chirality reversal at line defects crossing lithium tantalate crystals. Our results demonstrate a clear deviation from the ideal Ising configuration that is traditionally expected in uniaxial ferroelectrics, corroborating recent theoretical predictions of a more complex, often chiral structure.

[1] Université de Strasbourg, CNRS, Institut de Physique et Chimie des Matériaux de Strasbourg, UMR 7504, 67000 Strasbourg, France. [2] DQMP, University of Geneva, 24 Quai Ernest Ansermet, 1211 Geneva, Switzerland. [3] Department of Applied Physics, KTH—Royal Institute of Technology, Roslagstullbacken 21, 106 91 Stockholm, Sweden. Correspondence and requests for materials should be addressed to S.C.-H. (email: salia.cherifi@ipcms.unistra.fr).

Ferroic domain walls (DWs) have recently been shown to present unexpectedly rich and diverse physical properties beyond those of their parent materials. These include the observation of polar walls in otherwise non-polar materials[1], DW conductivity in insulating oxides[2–5], ferromagnetic ordering in ferroelectric antiferromagnets[6], and photovoltaic effects localized at the wall regions[7]. This plethora of unusual properties has transformed the perception of DWs in ferroelectric and multiferroic materials, which are now increasingly viewed as individual nanostructures with vast applicative potential in industry rather than as unwanted defects[8]. Similar to recent developments in nanomagnetism[9,10], DWs in ferroelectric and multiferroic materials are now considered as potential units of information in nanoelectronic device components[11,12].

In ferroelectrics, the physical properties of DWs are closely related to their internal structure. More specifically, they depend on the local symmetry[13]. By considering the point group symmetry of the domains and the changes of symmetry required to transform two adjacent domains, it can be deduced, for example, that a spontaneous electric polarization or magnetization is permitted at DWs in materials which do not display such ordering in their homogeneous form[14]. The local symmetry of DWs can also be deduced by considering the primary transition order parameter of the adjacent domains, and by taking into account a possible reduction of the parent order parameter space at the domain boundary region[15]. The early thermodynamic study by Lajzerowicz and Niez[16] predicting the existence of a phase transition within ferroelectric DWs has paved the way towards the investigation of unusual DW structures. Recent phenomenological and first-principles theoretical studies strongly indicate the existence of Néel or Bloch-type walls in uniaxial ferroelectrics[17–26], even in the simplest case of nominally uncharged 180° walls which were previously assumed to display Ising-type configuration.

Theoretical predictions concerning the non-Ising character of ferroelectric DWs are abundant in the recent literature and are largely accepted in the community. Nonetheless, the experimental observation of these predicted Néel and Bloch-type walls remains elusive. A key difficulty is the detection of a contribution arising from an extremely narrow region, since the DW width typically extends only over a few atomic cells. Recently, high-resolution transmission electron microscopy (TEM) has determined the atomic arrangement within ferroelectric domains[27] and revealed unconventional DW structures[28]. However, this technique is difficult to apply for general measurements due to its need for arduous sample preparation and its complex data analysis. A more frequently and more easily applied method to study the domain structure in ferroelectrics is piezoresponse force microscopy (PFM)[29]. In particular, vector PFM provides a 3D mapping of the polarization[30], which could in principle allow for a full description of the internal DW structure. However, in spite of its essential importance in probing ferroelectric domains, a downside of PFM is its inability to distinguish between an internal polarization structure and a possible electromechanical response arising from DW displacements or deformations. The latter effect is generally assumed to be a primary source for unusual signals obtained in DW studies[31]. Moreover, a solid interpretation of the PFM data requires that the measurements are combined with extensive theoretical modelling[32].

Nonlinear second-harmonic generation (SHG) microscopy offers a remarkable complementary approach, which not only allows for the non-perturbative observation of ferroelectric domains[33–35], but can also provide valuable insight into the 3D structure of DWs[36]. Obviously, the lateral resolution of SHG is lower than the resolution offered by TEM and PFM. Yet, the quadratic relation between the SHG signal intensity and the electric field amplitude of the fundamental wave provides high sensitivity to small local field discontinuities or enhancements, even if these effects are limited to a region with a size well below the lateral resolution limit[37]. A further advantage is that the nonlinear optical susceptibility tensor describing the SHG process is directly related to the symmetry of the system and to its ferroic order[35,38]. Therefore, by allowing to analyse the nonlinear susceptibility tensor elements, SHG measurements are particularly adapted for investigations of the point group symmetry and of the orientation of the domains (ref. 35 and references therein).

Here, through local SHG polarimetry measurements and numerical simulations exploiting symmetry considerations, we demonstrate the existence of a planar polarization component within the wall regions separating c-domains in uniaxial ferroelectric systems. This study is conducted mainly on tetragonal $Pb(Zr,Ti)O_3$ thin films (point group symmetry 4 mm). In addition, we investigate a trigonal $LiTaO_3$ bulk crystal (point group symmetry 3 m) as an independent test of our approach. In spite of significant differences between these two systems, a deviation from the idealized Ising configuration is demonstrated in both cases. While we observe Néel-type DWs in tetragonal $Pb(Zr,Ti)O_3$, periodically poled $LiTaO_3$ develops mainly Bloch-type walls. Depth-resolved SHG measurements conducted in the $LiTaO_3$ bulk crystal show that the Bloch-type walls are stable across the crystal depth. However, their chirality may change at line defects, in close analogy to Bloch lines in ferromagnets[39]. This study provides the experimental proof of a deviation from the conventional Ising-type configuration and shows the existence of chiral walls in ferroelectric materials. The functionalization of non-Ising DWs, in particular the switchable polarity of Bloch-type walls, represents promising features for the development of original nano-devices.

## Results

**Overview.** To prove the general applicability of our approach and its ability to identify a possible deviation of ferroelectric DWs from the ideal Ising-type configuration, we focus on nominally uncharged 180° domain DWs in two fundamentally different crystals: the one is a tetragonal lead zirconate titanate thin film, the other a lithium tantalate bulk trigonal crystal. This choice is motivated by theoretical studies predicting the existence of non-Ising walls with an internal structure specific to each system. A Bloch-type configuration has been predicted for Y-oriented walls in crystals of the lithium niobate family[40], while Morozovska[41] and Eliseev et al.[42] show that Néel-type DWs are expected in $PbZr_{0.2}Ti_{0.8}O_3$. Moreover, $PbZr_{0.2}Ti_{0.8}O_3$ is often used as a stand-in system for pure $PbTiO_3$ in which Néel-type DWs are expected[19,21].

**180° domain walls in tetragonal lead zirconate titanate thin films.** The internal structure of nominally uncharged 180° ferroelectric DWs is first investigated in a 50 nm-thick tetragonal $PbZr_{0.2}Ti_{0.8}O_3$ (PZT) layer in which c-domains have been patterned by means of a PFM tip (see Methods section for more details), as can be seen in Fig. 1b,c. The corresponding SHG image shows a localized emission at the domain boundary regions (Fig. 1d,e). The spectral analysis of the optical signal reveals an unambiguous SHG process by demonstrating a frequency doubling combined with a quadratic dependence of the intensity with the power of the fundamental wave (Fig. 1f–h). This result is particularly surprising since, using this measurement geometry with normal incidence and back reflection detection (Fig. 1a), and moreover, given the tetragonal symmetry of the film, no SHG signal should be generated from c-oriented domains (Supplementary Fig. 9). Besides, Ising-type DWs are

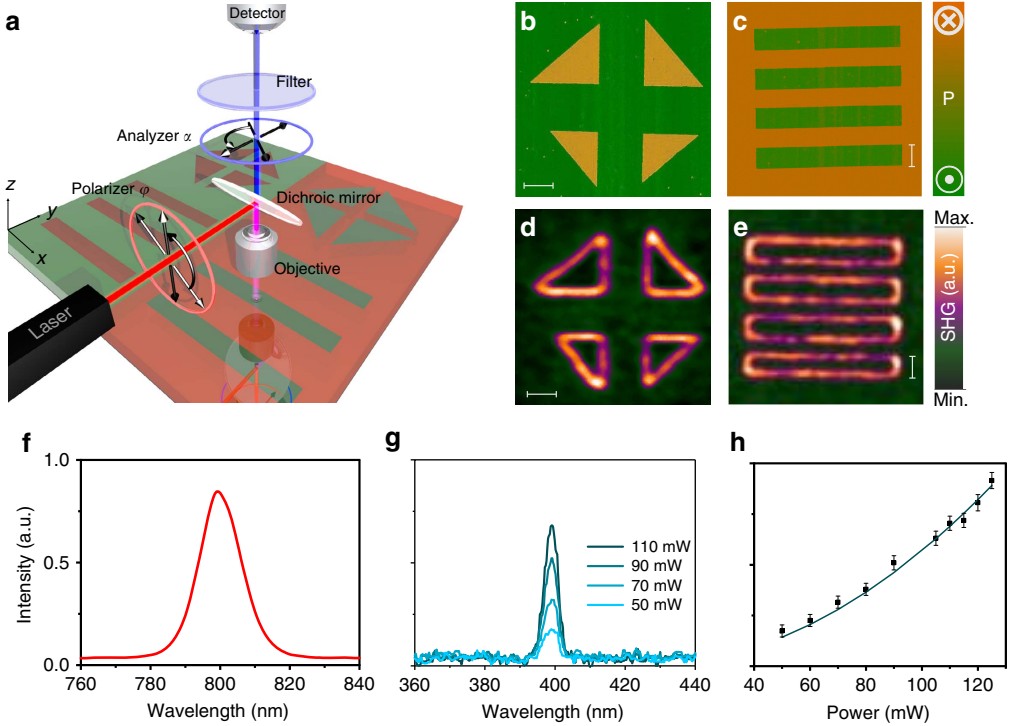

**Figure 1 | Detection of localized SHG at domain walls in PZT thin films.** (**a**) Schematic representation of the SHG experimental set-up. Ferroelectric domains ($c^+$ and $c^-$ domains) of different geometric shapes are written in the film by applying a bias voltage through a conductive PFM tip. The domain structure imaged by means of PFM is displayed for film regions with (**b**) right triangle and (**c**) rectangle-shaped domains. The corresponding nonlinear optical images reveal a localized optical signal at the walls surrounding the (**d**) triangular and (**e**) rectangular domains. The bars in all images correspond to a scale of 2 μm. (**f**) The fundamental wavelength is 800 nm while the localized emission (**g**) occurs at a wavelength of 400 nm, corresponding to the half of the fundamental wavelength. The scattered dots in **h** represent the variation of the emission intensity with the power of the fundamental wave. The error bars account for the intensity fluctuation of 10% and the continuous line is a quadratic fit of the experimental data (Supplementary Fig. 1 for more details). The SHG images represent the data recorded at a polarizer angle $\varphi = 0°$ integrated over the analyzer angles $\alpha$. The laboratory coordinate system ($x,y,z$) displayed in **a** coincides with the crystallographic axes ($X,Y,Z$) of tetragonal PZT.

centrosymmetric and they exhibit a vanishing polarization at the centre of the wall. Therefore, no SHG signal is expected at the domain boundary regions either in the case of an Ising-type DW.

Earlier studies based on observations by means of far field, near field and Cherenkov-type SHG[43–45] have shown that the domain boundary regions can appear either as dark lines[34,46,47], due to destructive interference of the SHG between opposite domains (out-of-phase), or as bright regions[43,48,49]. Although SHG signals at DWs have been reported in the studies mentioned above, a thorough understanding of the origin of the SHG emission remains elusive[48]. The effect was first attributed to structural defects[43] agglomerated at the DWs, and to the resulting loss of coherence[49]. However, the detection of Cherenkov radiation at the DWs demonstrates the coherence of the generated light. The role of the defects was then dismissed in favour of other contributions such as localized d.c. electric fields induced by strain[46]. In this context, we conduct a precise analysis of the SHG signal of the DWs by means of polarimetry measurements and numerical simulations based on symmetry arguments. We conclude that the SHG signal at PZT and LiTaO₃ DWs is not a spurious effect, but a result of the particular internal polarization structure of the DWs.

Modelling the SHG signal for Néel or Bloch-type walls requires the knowledge of the related nonlinear optical susceptibility tensor (that is, the $d_{ij}$ elements of the SHG tensor). Assuming that the optical tensor is isomorphic to the piezoelectric tensor, the same transformation operations which hold for the piezoelectric tensor at a given symmetry also apply to the SHG tensor. Any new or modified tensor corresponding to an arbitrary coordinate

system with a specific ferroelectric polarization of the underlying material can thus be deduced using the rotation transformation relations as they are commonly applied to piezoelectric tensors[50]. For this, it is sufficient to know the reference SHG tensor $d^0$ defined in the crystallographic reference frame of the parent phase. A detailed description of the transformation method as well as the resulting SHG tensors obtained in Néel and Bloch-type DWs is provided as Supplementary Note 2. Knowing the susceptibility tensors, the SHG intensity can be calculated for any wall type as a function of the polarizer angles ($\alpha$ and $\varphi$) using the analytic form given in the Methods section (equations (1) and (2)).

The two-dimensional (2D) simulation of the SHG is obtained by subdividing the DWs into discrete regions, in which the ferroelectric polarization is allowed to rotate, for instance in horizontal DWs ($\parallel x$-axis), within the ($zy$)-plane in the case of Néel-type walls, and in the ($zx$)-plane in the case of Bloch-type walls. The SHG intensity is calculated at each rotation angle assuming a small amount of experimental noise, that is, random fluctuations of the polarization angle ($\pm 0.5$ rad) around its equilibrium position. Fig. 2d,e displays simulated SHG images obtained for square-shaped c-oriented domains, with horizontal (HDWs $\parallel x$-axis) and vertical (VDWs $\parallel y$-axis) walls aligned with the cubic crystallographic axis of PZT (the laboratory coordinate system ($x,y,z$) coincides with the crystallographic axes ($X,Y,Z$)). The SHG images are simulated at different settings of the analyzer and polarizer angles, in the case of Néel-type (Fig. 2d) and Bloch-type DWs (Fig. 2e). A clear SHG signal is observed at the DW regions in both configurations. However, the polarimetry

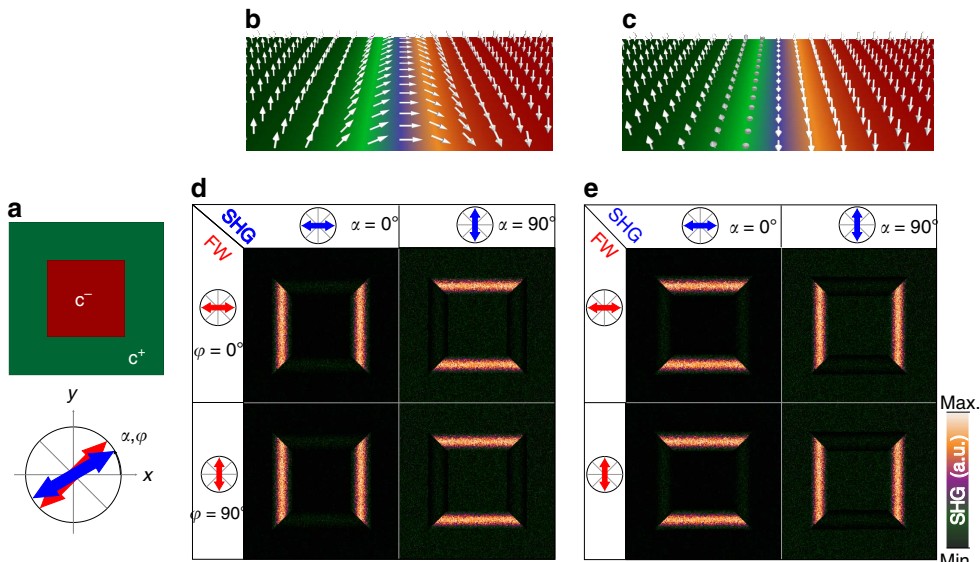

**Figure 2 | 2D simulations of the SHG emission at PZT c-domain boundaries.** The numerical simulations of the SHG emission are performed for square-shaped c-domains in tetragonal PZT as outlined in **a**, assuming DWs with (**b**) Néel-type and (**c**) Bloch-type internal structure. The variation of the SHG intensity at horizontal (HDWs || x-axis) and vertical (VDWs || y-axis) DWs is displayed in the case of (**d**) Néel and (**e**) Bloch walls at different polarizer and analyzer angles ($\varphi$ and $\alpha$, respectively) taken with respect to x-axis (schematically represented by red and blue arrows).

selection rules are modified, depending on the considered wall type. According to the simulations, Néel-type DWs should solely be observed when the SHG analyzer axis is oriented perpendicular to the walls ($\alpha = 0°$ in case of VDWs and $\alpha = 90°$ in case of HDWs), while Bloch-type DWs are visible when the analyzer axis is along the walls (the analyzer orientation is indicated by the blue arrow in Fig. 2d,e). These results indicate a preferential SHG polarization perpendicular to the walls in the case of Néel-type configuration and parallel to them in the case of Bloch-type DWs. When the polarization of the fundamental beam is turned from $\varphi = 0°$ to $\varphi = 90°$, only the SHG intensity varies at horizontal and vertical DWs and not the anisotropy of the signal.

To experimentally probe the internal structure of the walls and their possible transformation depending on their orientation with respect to the crystal, we consider domains with rectangular shape in which the walls are parallel to the cubic axes X and Y of the crystal, as well as quasi-isosceles right triangle domains containing oblique walls (Fig. 1b,c). Figure 3c,d shows the corresponding SHG images measured at different analyzer and polarizer settings. While HDWs and VDWs present in rectangles and at the short sides of the triangles follow polarimetry selection rules similar to those simulated in Néel-type DWs (Fig. 2d), the oblique domain walls (ODWs 1 and 2) at the hypotenuse seem to exhibit a different behaviour. Their polar plots (magenta and violet scattered points in Fig. 3f) show that the lobe maximum of the SHG is roughly along the oblique walls, with an uncertainty of $\pm 5°$. This is in contrast to VDWs and HDWs, where the SHG polarization maximum is found perpendicular to the walls (orange-scattered points in Fig. 3e,f).

Further insight into the internal structure of the walls in tetragonal PZT is obtained through the theoretical fitting of SHG polarimetry data measured at horizontal, vertical and oblique walls (Fig. 3e,f). The simulations are based on the analytic derivation of the SHG signal, taking into account the symmetry of the parent material and the orientation of the wall with respect to the crystal axes. In these simulations, we assume that a non-zero average planar component of the ferroelectric polarization exists in the DW regions. The results are compared by assuming an internal polarization that is either parallel (Bloch) or perpendicular (Néel) to the walls (Supplementary Fig. 8). The polarimetry analysis of the SHG signal arising from HDWs, VDWs as well as ODWs (scattered orange dots in Fig. 3e,f) fits well with the analytic model accounting for a Néel-type configuration (continuous line in Fig. 3e,f).

A comprehensive simulation of the SHG at 180° DWs is performed based on the symmetry of the parent phase, the orientation of the walls with respect to the crystal, and their hypothetical internal structure. Figure 3g shows the three-dimensional (3D) representation of the SHG polarimetry simulated at HDWs, VDWs and ODWs1–2 in the case of a Néel-type configuration. ODWs are placed at $+45°$ (ODW1) and $-45°$ (ODW2) with respect to x-axis. Different visualization angles are provided in Supplementary Movie 1. This 3D visualization allows for the full representation of the SHG polarimetry from which polar plots can be extracted through plane cuts at constant $z = \varphi$. The experimental polar plots measured at $\varphi = 0°$ (scattered dots in Fig. 3e,f) can thus be related to the simulated plots displayed in Fig. 3h. Note that due to the experimental difficulty to account for absolute SHG intensities, the normalized SHG intensities are displayed. By doing so, the SHG anisotropy can be identified immediately. As a result, any comparison between the experimental and the simulated data shown here should only refer to the shape and orientation of the polar plots, and not the absolute SHG intensity. A clear azimuth angle variation of the SHG polarization maxima is evidenced depending on the orientation of the walls. In agreement with the experiments, the maximum SHG polarization in the Néel-type configuration is found perpendicular to VDWs and HDWs, while ODWs show quasi-axial SHG lobes. Nonetheless, an angular deviation of about 15° is observed between the experiments and the simulations in the case of oblique walls. This difference cannot be explained solely by a misalignment of the wall in the experiments ($\pm 5°$). A modification of the nonlinear optical tensor elements of up to 45% does not affect the SHG polarization in the case of HDWs and VDWs, whereas angle variations larger than 20° can be found for ODWs (Supplementary Fig. 11). Therefore, local variations of the nonlinear optical tensor

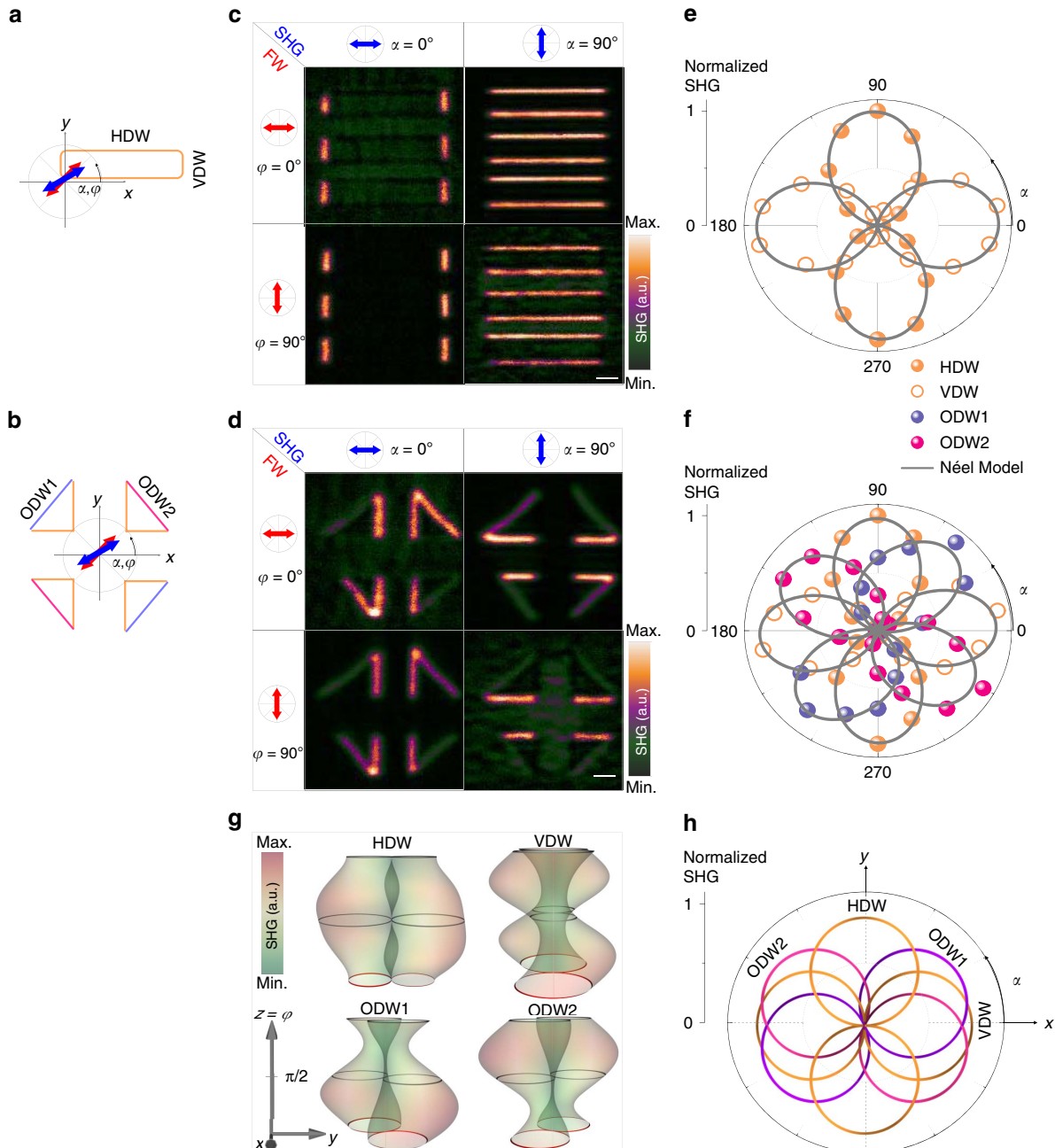

**Figure 3 | Polarimetry analysis of the DWs' SHG signal in tetragonal PZT.** A systematic analysis of the local SHG is conducted for different polarizer and analyzer settings, at horizontal (HDW), vertical (VDW) and oblique (ODW) domain walls, as schematically illustrated in **a,b**. SHG measurements conducted in **c** rectangular-shape c-domains exhibit maximum signal at HDWs and VDWs when the analyzer (blue arrow) is perpendicular to the walls. The same result is observed at HDWs and VDWs in **d** triangular-shape c-domains, while oblique walls referred to as ODW1 and ODW2 exhibit a different behaviour. The corresponding polar plots (scattered dots) of the normalized SHG intensity measured are shown in case of (**e**) rectangles and (**f**) angle triangles at a fundamental polarization angle $\varphi = 0°$. The continuous lines are fits of the experimental data to the analytic expression of the SHG intensity expected for Néel-type walls with horizontal, vertical or oblique orientations. (**g**) 3D SHG polarimetry simulated in the case of a planar ferroelectric polarization along the $y$-axis (Néel-type), represented in cylindrical coordinates ($x = I^{SHG} \cos \alpha$, $y = I^{SHG} \sin \alpha$ and $z = \varphi$). The z-axis has been scaled with a factor 10 to allow for a better visibility. The colour map represents the SHG intensity in a.u. and the continuous lines result from plane cuts at $\varphi = 0°$ (red colour), $\varphi = 90°$ and 180° (grey). The normalized intensity of the calculated polar plots at $\varphi = 0°$ is displayed in **h** to facilitate the comparison with the experimental results.

elements may be at the origin of the observed difference between the calculations and the experiment at ODWs.

**180° domain walls in lithium tantalate bulk crystals.** For comparison, the polarimetry analysis of the SHG signal is similarly conducted in 500 µm-thick periodically poled nearly

stoichiometric LiTaO₃ (LT) bulk crystals. In this system, c-domains forming either micrometric stripes or hexagons have been produced by means of electric field poling (see Methods section for more details). To suppress spurious surface contributions to the SHG signal, the measurements were performed at a focal distance of 100–300 µm below the top

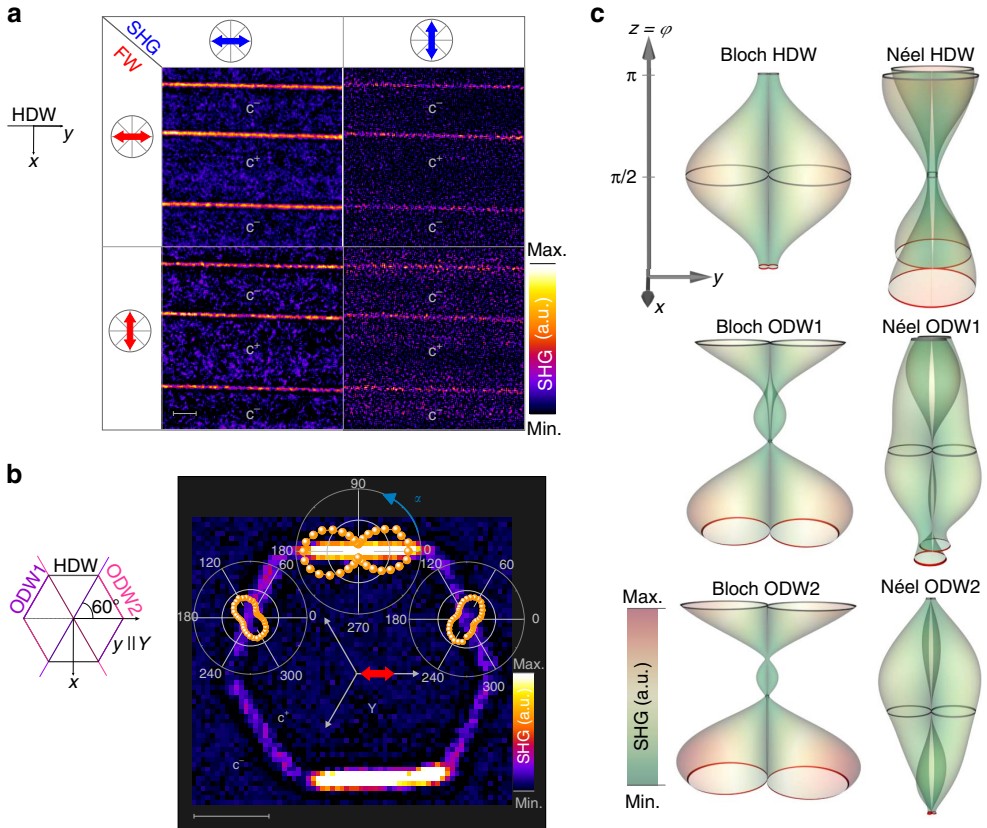

**Figure 4 | SHG Polarimetry analysis in LiTaO₃.** SHG images of (**a**) alternated c⁺ and c⁻ stripe domains and (**b**) typical hexagonal c-domains in periodically poled nearly stoichiometric LiTaO₃ measured at different polarizer ($\varphi$) and analyzer ($\alpha$) angles. The polar plots measured at the boundary regions of the hexagon (orange dots in **b**) show a maximum SHG polarization along the three equivalent directions Y of the crystal [11–20], [1–210] and [−2,110]) represented by grey arrows. The bars in all images correspond to a scale of 5 μm and the colour map in **a,b** refers to the bulk SHG. (**c**) The 3D SHG polarimetry has been calculated assuming the existence of a planar ferroelectric polarization oriented either parallel (Bloch) or perpendicular to the walls (Néel). The simulation results are displayed both for horizontal (HDW || Y-axis) and oblique walls (ODW1 and ODW2 at ± 60° with respect to Y-axis). The colour map denotes the normalized SHG intensity in a.u., and the continuous lines represent the SHG polar plots at $\varphi = 0°$ (red colour) $\varphi = 90°$ and 180° (grey).

surface. Figure 4a displays a clear SHG signal at the boundary regions of the alternated stripe domains. As opposed to PZT, the SHG polarimetry analysis shows maximum SHG polarization along the walls (|| y) in LT, regardless of the fundamental wave polarization. Nevertheless, a small SHG signal is still measurable when the analyzer angle of the SHG polarization is perpendicular to the walls. This result is clarified in the following analysis of the SHG polarimetry based on symmetry arguments, in particular at oblique walls in hexagonal domains.

C-domains with hexagonal shape patterns are obtained in LT under stoichiometric conditions as described in the Methods section. The domains are delimited by DWs along the three equivalent Y-crystallographic directions (represented by grey arrows in Fig. 4b). A typical SHG image of a hexagonal c-domain is displayed in Fig. 4b. Localized nonlinearities are clearly visible at all six equivalent Y-DWs of the hexagon. However, a larger SHG intensity is detected in HDWs (|| y-axis) when the SHG polarization is along y-axis ($\alpha = 0°$), while oblique walls (ODW1 and 2) show maximum SHG polarization at +60° with respect to y-axis in the case of ODW1, and at −60° with respect to y-axis in the case of ODW2. As previously explained in the case of PZT, the SHG response depends sensitively on the probed nonlinear optical coefficients of the system. Calculations accounting for the local symmetry at the walls and their orientation are provided for trigonal LT to better clarify the SHG polarimetry response (Fig. 4c and Supplementary Movie 2). A comparison of the

experimental results (Fig. 4a,b) with simulated polar plots (continuous lines in Fig. 4c) shows that in the case of trigonal LT the SHG signal at the DWs is consistent with a dominant Bloch-like configuration. On the other hand, due to the large imbalance between the nonlinear optical coefficients in the case of trigonal LT (Methods section), the SHG intensity is strongly affected by the DW types and their relative orientation with respect to the photon analyzer and polarizer angles. For example, the simulated SHG signal for horizontal Néel walls shows a low intensity as compared to the intensity expected in the case of Bloch-type wall (see the simulated 3D SHG displayed in Fig. 4c and the corresponding colour map). This means that a potential Néel component would hardly be detectable. Nevertheless, and in spite of its small value, the SHG signal related to a Néel-type component could be detected when the analyzer angle was perpendicular to the walls (right-hand column in Fig. 4a). This finding is in perfect agreement with the theoretical predictions of Lee et al.[19,20] concerning the mixed Bloch-Néel nature of 180° DWs in trigonal lithium niobate.

The observation of Néel or Bloch-type configurations in these materials does not imply the absence of an Ising component. In uniaxial ferroelectrics, the Ising character should dominate due to electrostrictive and electrostatic effects. Therefore, a perfect Néel or Bloch-like rotation of the polarization vector is unlikely. Instead, a coexistence of Ising-Néel or Ising-Bloch configurations is expected[19]. In this case, the amplitude of the polarization

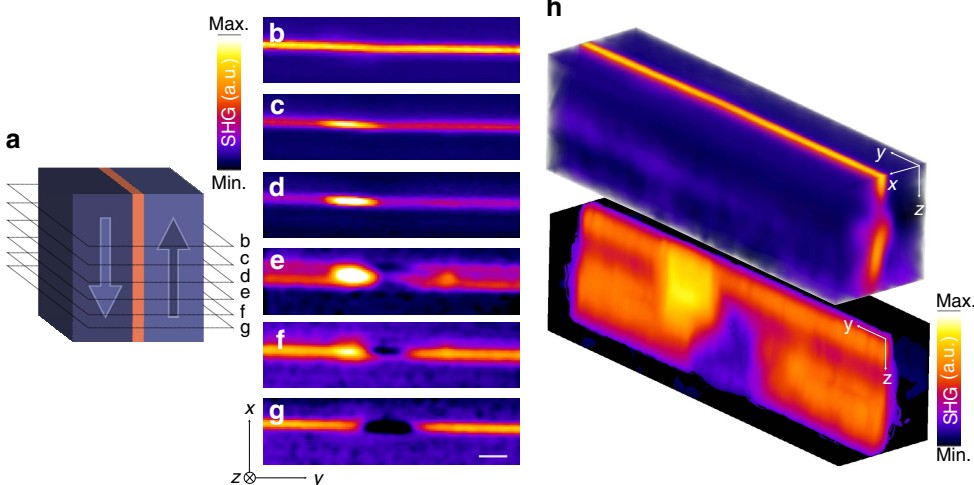

**Figure 5 | Depth profile of a Bloch-type chiral wall in LiTaO₃.** (**a**) Schematic three-dimensional representation of a $z$-oriented periodically poled bulk LiTaO₃ crystal with stripe domains. The plane cuts represented by dotted lines correspond to the laser focus position along $z$-axis corresponding to the (**b**–**g**) 2D SHG datacube maps measured in the ($xy$) plane. The images are recorded at different depths ranging from (**b**) $z = 300\,\mu m$ up to (**g**) $z = 400\,\mu m$ in steps of about 15 μm. Here $z = 0$ corresponds to the top surface. The bar corresponds to a scale of 5 μm and it is common to **b**–**g**. A tomographic reconstruction showing the 3D representation of the Bloch-type DW is shown in (**h**). All the images are recorded at $\varphi = 0°$ and $\alpha = 0°$. The colour map represents the bulk SHG intensity in a.u. measured in a sample volume of $12 \times 55 \times 100\,\mu m^3$.

decreases during the rotation and it vanishes at the centre of the wall. The possibility of opposite rotation of the polarization at each side of the wall has also been predicted[51]. We refer to this type of mixed walls as bichiral DWs, following the terminology used by Houchmandzadeh *et al.*[52]. Our method cannot distinguish between pure forms of Néel (Bloch) DWs and mixed variants thereof, such as a mixed Ising-Néel or Ising-Bloch configurations (Supplementary Discussion) or bichiral mixed walls. Even though SHG microscopy does not provide the spatial resolution required to image the precise rotation of the polar vector at the atomic scale, this study shows that dominating Bloch or Néel-type configurations can be revealed and discriminated by means of local SHG polarimetry.

**Bloch lines in periodically poled lithium tantalate.** Bloch-type walls may exist in two variants since the ferroelectric polarization between c-domains of opposite sign can rotate either clockwise or counterclockwise across the wall. The absence of mirror symmetry makes these walls chiral. Their chirality combined with its possible reversal by moderate external fields propels Bloch walls to the first position among the most promising active nanofunctional components for memory devices[53]. To explore the chirality of Bloch-type walls in periodically poled LT, we focus on the evolution of the 3D internal structure of the walls by means of SHG tomography. Figure 5 displays the depth-resolved SHG signal from the domain wall region. The bright area in Fig. 5b is the most common wall configuration observed in periodically poled LT crystals, and it is typical for a homochiral Bloch-type transition. However, intensity variations can be observed along the domain wall, which moreover change along the depth (Fig. 5b–g). While the spatial resolution and sensitivity of the experimental method is not sufficient to unambiguously identify the 3D arrangement of the polarization field in the domain wall region, it is possible to interpret some of the observed features with a good degree of confidence. In particular, the dark regions strongly suggest the presence of Bloch Lines[39]. In ferromagnetic systems, a Bloch Line defines a narrow region separating Bloch walls with opposite chirality. It has been predicted that such domain wall structures should exist also in ferroelectrics[53,54]. Although SHG microscopy does not allow for

the discrimination between DW segments with antiparallel internal polarization (because of its quadratic dependence on the susceptibility), a Néel type transition between the two segments should result in the SHG image as a low-intensity region. Since we have repeatedly observed such variations within domain walls (Supplementary Fig. 6c), we conclude that they are genuine features of the polarization structure. All the measured cross-sections show such a contrast variation pointing to a Bloch line within the domain walls. At the vicinity of Bloch lines, we observe both a small wall distortion and an increased SHG intensity. Moreover, the pronounced modulation of the SHG signal along the depth (see ($yz$) view in Fig. 5h) suggests an even more complex 3D wall structure, which is reminiscent of topological features known from micromagnetism. Our observation of the 3D contrast variations demonstrates that the polarization in the domain wall region is highly non-trivial, although it is predominantly of Bloch type. This signal may result from local chiral structures of the polarization within the domain wall triggered by local strain field distortions.

## Discussion

Nonlinear optical microscopy is used to probe the internal structure of ferroelectric domain boundaries. The non-Ising character of nominally neutral 180° DWs is demonstrated in two different systems. Néel-type DWs are observed in tetragonal PZT thin films and a dominant Bloch-type configuration is evidenced in quasi-stoichiometric LT crystals. This work shows that the combination of local polarimetry SHG and numerical simulations based on symmetry arguments constitutes a valuable tool for mapping polarization components at ferroelectric DWs.

Little is known so far about the origin of the non-Ising structure of DWs. This deviation could be explained by localized electromechanical coupling. For example, lattice distortions caused by oxygen octahedron rotations in perovskite oxides are capable of exhibiting improper ferroelectricity (for example, at elastic twins) due to the coupling of the flexoelectric effect (polarization induced by strain gradient) and rotostriction[55]. However, this effect should play a minor role in uniaxial ferroelectrics like those investigated in this study. Only two theoretical studies that we know of have investigated the origin of

the deviation of nominally uncharged 180° DWs from the Ising-type model[51,56]. These studies have shown that the domain wall type in tetragonal BaTiO$_3$ is determined by the competition between the depolarization and the flexoelectric fields.

We hope that our study will stimulate further experimental and theoretical interest in chiral walls. In particular, the study of strain-gradient variations could reveal exciting patterns and distinct properties that could be useful for the design of nanoferroic devices based on domain walls.

## Methods

**Pb(Zr,Ti)O$_3$ film preparation.** Tetragonal PbZr$_{0.2}$Ti$_{0.8}$O$_3$ (50 nm) films (PZT) are epitaxially grown on SrRuO$_3$(35 nm)-buffered SrTiO$_3$(001) single crystals by means of off-axis radio-frequency magnetron sputtering, following the deposition route described in ref. 57. A single domain state (c$^-$) is obtained with a uniform polarization along the z-axis, out of the plane of the film. Domains of opposite polarization separated by nominally neutral 180° DWs are prepared by applying a bias voltage ($\pm$ 10 V) between a conductive PFM tip and the SrRuO$_3$ base electrode. The PFM tip is scanned over a selected area to pattern the desired domains.

**Domain engineering in LiTaO$_3$.** We used commercial (Oxide Corp) monodomain 0.5 mm-thick Z-cut 1 mol% MgO-doped nearly stoichiometric LiTaO$_3$ crystals. Hexagonal bulk domain structures with 180° DWs aligned with the Y-crystallographic axes were fabricated by electric field poling, through the application of 1.8 kV voltage pulses of 80 ms duration via patterned metallic (Ti) electrodes deposited on the $+Z$ face[58], and an uniform electric (gel) contact on the opposite side ($-Z$). The individual hexagonal domain shape matched the preferential domain wall orientations observed in the material (six equivalent walls along the Y-axis, that is, [11−20], [1−210] and [−2,110])[59]. After the poling procedure, the metallic electrodes were removed by a wet-etching step in a 48% HF:H$_2$O solution, followed by a thorough ultrasonic cleaning in organic solvents. The same method was used to fabricate 1D arrays of 2 mm-long stripe domains aligned to the Y-axis, with a periodicity of 25 μm along X.

**Second-harmonic generation measurements.** Local SHG measurements are conducted by means of a scanning confocal microscope. The fundamental wave is provided by a Spectra Physics Ti:Al$_2$O$_3$ laser (Millenium-Tsunami combination), which generates 100 fs pulses with a repetition rate of 80 MHz and a wavelength centreed at 800 nm. The laser beam is directed at normal incidence to the sample, and focused with a × 40 magnification objective lens (numerical aperture N.A. = 0.66). The SHG images are obtained by scanning the sample with respect to the incoming beam using computer-controlled stepping motors with a minimum step of 100 nm, and recording the SHG signal at each scan step with a typical exposure time is of 20–40 ms per step at a power of 100–150 mW. The output intensity was spectrally filtered and collected into a photomultiplier box. Polarimetry measurements are performed by recording the images at different polarizer and analyzer angles ($\varphi$ and $\alpha$, respectively). See Supplementary Note 1 for more details about the data acquisition and processing procedures.

**SHG modelling.** The SHG emission involves the coupling of two incident photons at frequency ω to produce a dipole characterized by a polarization $\mathbf{P}^{2\omega}$ oscillating with the double frequency. The leading term in the frequency doubling process is the second-order nonlinear optical susceptibility tensor $\chi^{(2)}$. The elements of the susceptibility tensor are usually replaced by a contracted d-tensor, following the Voigt notation (contracted tensor elements $d_{ij}$, $i = 1$–3 and $j = 1$–6). $\mathbf{P}^{2\omega}$ depends on the optical susceptibility tensor ($d_{ij}$ elements) and on the electric field of the fundamental wave $\mathbf{E}(E_x, E_y, E_z = 0)$ as follows:

$$\begin{pmatrix} P_x^{2\omega}(\varphi) \\ P_y^{2\omega}(\varphi) \\ P_z^{2\omega}(\varphi) \end{pmatrix} = \varepsilon_0 \begin{pmatrix} d_{11} & d_{12} & d_{13} & d_{14} & d_{15} & d_{16} \\ d_{21} & d_{22} & d_{23} & d_{24} & d_{25} & d_{26} \\ d_{31} & d_{32} & d_{33} & d_{34} & d_{35} & d_{36} \end{pmatrix} \begin{pmatrix} E_0^2 \cos^2 \varphi \\ E_0^2 \sin^2 \varphi \\ 0 \\ 0 \\ 0 \\ E_0^2 \sin 2\varphi \end{pmatrix} \quad (1)$$

where $\varphi$ is the polarization angle of the fundamental wave. The complete polarimetry representation of the SHG polarization as a function of the polarizer and analyzer angles $\varphi$ and $\alpha$, respectively, is obtained using the Jones formalism:

$$\begin{pmatrix} P_x^{2\omega}(\varphi, \alpha) \\ P_y^{2\omega}(\varphi, \alpha) \\ P_z^{2\omega}(\varphi, \alpha) \end{pmatrix} = \begin{pmatrix} \cos^2\alpha & \cos\alpha \sin\alpha & 0 \\ \cos\alpha \sin\alpha & \sin^2\alpha & 0 \\ 0 & 0 & 1 \end{pmatrix} \begin{pmatrix} P_x^{2\omega}(\varphi) \\ P_y^{2\omega}(\varphi) \\ P_z^{2\omega}(\varphi) \end{pmatrix} \quad (2)$$

Equations (1) and (2) allow for a complete description of the SHG intensity $I^{SHG}(\varphi,\alpha) \approx |\mathbf{P}^{2\omega}(\varphi,\alpha)|^2$, provided that the symmetry of the material (that is, the $d_{ij}$ elements of the susceptibility tensor) is known, and vice versa. By choosing the set of polarizations ($\varphi$ and $\alpha$ angles) properly aligned with respect to the crystal orientation, the elements of the $\chi^{(2)}$ tensor can be directly accessed through

polarimetry measurements (polar plots). Quantitative evaluations tend to be difficult as they require relative measurements with respect to a reference system (for example, quartz or ammonium dihydrogen phosphate) and specific measurement geometry. Nevertheless, the qualitative probe of the $\chi^{(2)}$ elements provides valuable information on the symmetry and on the ferroic domain structure.

SHG polarimetry data (polar plots) were fitted at a given polarization of the fundamental wave ($\varphi$ constant) using the analytic model deduced from equations (1) and (2) by assuming various DW types. The agreement with the data or lack thereof allowed us to uniquely infer the structure of the observed DWs. The susceptibility tensor in equation (1) is specified in Supplementary Tables 1 and 3 for each domain wall type and geometry (horizontal, vertical and oblique walls).

The 2D simulations of the SHG images were obtained by subdividing the DWs into discrete regions, in which the ferroelectric polarization is allowed to rotate. The $\chi^{(2)}$ SHG tensor is calculated at each rotation angle. The result is then inserted as input parameter in equations (1) and (2). This allows the calculation of the SHG intensity at any position in domains and domain wall regions for given polarizer and analyzer angles. The numerical value of the SHG tensor elements used in the simulations are $d_{31} = -11.09$ pm V$^{-1}$, $d_{15} = -11.09$ pm V$^{-1}$ and $d_{33} = -18.34$ pm V$^{-1}$ for tetragonal PZT[60] and $d_{31} = 1.54$ pm V$^{-1}$, $d_{22} = 0.46$ pm V$^{-1}$ and $d_{33} = 12.9$ pm V$^{-1}$ in trigonal LT[61]. The SHG intensity is calculated by taking $\varepsilon_0 E_0 = 1$ and it is presented in arbitrary units.

**Data availability.** The data that support the findings of this study are available from the corresponding author upon reasonable request.

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

## Acknowledgements

This work was supported by the French National Research Agency (ANR) through JCJC program (DYNAMECS, ANR-11-JS10-009-01). K.G. acknowledges support from the Swedish Research Council (ADOPT Linnaeus Centre, grants VR 622-2010-526 and 621-2011-4040). P.P. acknowledges financial support from the Swiss National Science Foundation through the NCCR MaNEP and Division II grant 200021-153174. Helpful discussions with O. Crégut (IPCMS) on the SHG measurements are acknowledged.

## Author contributions

S.C.-H. designed the study and directed the project. The SHG measurements were performed by G.T. and S.C.-H., in collaboration with K.D.(H.)D. The 2D simulations of the SHG have been conducted by H.B. with feedback from S.C.-H. The data processing procedures and the 3D simulations were developed and conducted by S.C.-H. and R.H. The ferroelectric domain lithography in thin PZT films and the related PFM measurements were performed by J.G., I.G. and P.P. The 1D (strips) and 2D (hexagons) periodic domains were poled in LiTaO₃ ferroelectric photonic crystals by K.G. S.C.-H. wrote the manuscript with contributions from R.H., K.G. and P.P. All authors discussed the experimental results and models, commented on the manuscript and agreed on its final version.

## Additional information

**Competing interests:** The authors declare no competing financial interests.

**Publisher's note**: 

