## [Peer Review File · Nature Communications]

Reviewers' comments:

Reviewer #1 (Remarks to the Author):

This paper focused on the local symmetry of ferroelectric domain walls for two classical ferroelectric materials, i.e., PZT and LiTaO₃. The authors provided new evidences to support non-ising domain walls existing in ferroelectrics by the SHG method. The results are important for the fundamental understanding of ferroelectrics and the design of potential domain-wall based nano-devices. Before the manuscript can be accepted for publications, the following concerns should be addressed.

1. SHG method is used to determine the symmetry of domain walls, but it is not able to map the detailed polar vectors within domain walls. Therefore, it is difficult to determine the type of domain walls (i.e., Neel, Bloch, etc.) by the SHG method alone. The authors determined the types of domain walls by modeling the SHG signals. The question is that: Is there a one-to-one correspondence between the domain-wall types and SHG signals in modeling? Is it possible that two or more domain-wall types (may not be Neel or Bloch) can generate a similar or the same SHG signal?

2. Based on Figure 1 and 5, it appears the thickness of domain walls is in the range of 100~1000 nm. This value is very large compared to those determined from TEM observation which show that the thickness of domain walls is generally around several nanometers [e.g., Jia CL et al, Nat Mater, 7, 57-61 (2008)]. Is it due to the low resolution of SHG?

3. For PZT thin films, the authors designed two artificial domain shapes by using a PFM tip, where non-ising domain walls were observed. However, these two domain shapes cannot be achieved by naturally. Are the types of domain walls highly related to the geometric shapes of the domains? Is it possible to observe a non-ising domain wall in naturally formed domain patterns in PZT? To answer this question, the authors need to do SHG experiments on a PZT thin film without written domains.

4. The origin of non-ising domain walls is not clearly discussed in the present paper. The authors need to give a specific discussion on the origin of non-ising domain walls for each cases (i.e., PZT and LT). The Ising-like domain walls should also exist in PZT and LT for certain conditions. So, which conditions favor Ising-domain walls and which favor Neel or Bloch domain walls? This issue is important for the design of ferroelectric devices based on domain walls.

Reviewer #2 (Remarks to the Author):

This paper describes observations of domain walls (DWs) in a (PbZr)TiO₃ (PZT) film and a LiTaO₃ (LT) single crystal obtained using a reflection-type SHG microscope (SHGM). The research point is that the authors fabricated 180 domain structures patterned by the PFM and observed them. They claim that the DWs of PZT are of the Neel type, while those of LT are mainly of the Bloch type but mixed with Neel type. It is the first study on the chirality of DWs using SHGM, and the experimental results are reliable and interesting for fundamentals and applications. However, the analyses of the experimental results are too simple and need more precise considerations for getting final conclusions. At least, the following points should be fully taken into account.

(1) The authors consider only Ps chirality and do not think over the oxygen octahedron rotations (OORs), which have been pointed out to play the important role at DWs in some perovskite ferroics, as discussed in the references below. The OOR could cause the symmetry lowering at DWs. The authors should examine the effect and make comments on the possibility of this kind of symmetry lowering when they analyze the SHG anisotropy.

1-1. Mark Calleja et al. J. Phys.: Condens. Matter 15 (2003) 2301.

1-2. Anna N. Morozovska et al. PHYSICAL REVIEW B 85, 094107 (2012).

1-3. S. Van Aert, et al. Adv. Materials 24, 523 (2012).

(2) In this research, the reflection-type SHGM is used. However it is well known that the analysis of this type of experiments is very complicated, because various factors affect the results (see the references below). For example, are SH waves generated when the fundamental waves (FWs) go forward in the sample, or they go back after being reflected at the substrate (PZT) or the bottom surface (LT)? The former is the inverse coupling case with the wave number mismatch, $\Delta k_i = k(2\omega) + 2k(\omega)$, and the latter is the regular coupling case, $\Delta k_r = k(2\omega) - 2k(\omega)$. The two cases give different behavior of SH waves. The authors should discuss this problem in the manuscript.

2-1. E. Sipe et al. PRB, 35, 9091 (1987).

2-2. S. W. Liu et al. JAP, 101, 104118 (2007).

(3) It has been also pointed out that the internal observation of the sample using the SHGM depends strongly upon the sign of Δk . (see the reference below). In the case of Δk positive, SHG intensities from the inside are very weak, which makes the observation difficult. In LT crystals, all SHG tensor components are known to be positive and a special care is necessary for the observations. The authors should make some comments on it.

3-1. J. Kaneshiro et al. J. Appl. Phys. 104, 054112 (2008).

(4) What happens at the boundaries of the x-axis elongated DW and the y-axis elongated one? (Fig. 1(e) for example).

(5) What factors determine the type of DWs, Ising type, Néel type, or Bloch type?

(6) For observations of DWs in LT, the domain structure was patterned using the metal electrodes, and then the electrodes were removed by the chemical etching technique. In this case, it is expected that DWs are topologically different from the up and down domains. Does this affect the SHG intensities?

(7) In Fig.1, the directions of P and A are not so clear. It should be noted in the caption.

(8) In the caption of Fig.1, "lead titanate" should be "PZT".

Reviewer #3 (Remarks to the Author):

Authors claim that they have obtained experimental evidence of the predictions on the non-Ising configuration of ferroelectric domain walls. Not only that, they say that they demonstrate the predicted chirality of some of these walls. It is interesting that they obtain this information from a low lateral resolution technique as Second Harmonic Generation (SHG). In my opinion, these results are of high relevance for the ferroelectrics community, and in general for all researchers working with ferroic materials. The properties of these materials strongly depend on the nature of ferroic walls, and, furthermore, these walls have recently become themselves the focus of attention. This work may contribute to new advances in this field.

Authors only analyze two cases. First, Lead Zirconate Titanate (note that the word "zirconate" is missing in some parts of the text: abstract, figure 1 caption...) thin films. And, secondly, Lithium Tantalate single crystals. The choice of these specific materials is not clearly stated in the text, but it seems that the PZT film is only used to show the experimental demonstration of the non-Ising nature of ferroelectric domain walls. In this case authors fail to cite any previous calculations on domain walls that predict their Néel-type character in tetragonal PZT. Probably these references do not exist, but then, why choose tetragonal PZT if not previous simulations were reported? Being a thin film, it is a pity that we do not have the opportunity to see the 3D study of the domain wall, similar to the one performed for LT single crystal. In this second case we can compare the experimental results with theoretical predictions. In my opinion, authors should discuss the choice made and the applicability of this method to the study of the nature of ferroelectric walls in other compositions and materials in order to strengthen the conclusions drawn.

Another interesting factor is the fact that none of the measurements at a local scale as Transmission Electron Microscopy or Piezoresponse Force Microscopy has been able to unveil the character of these walls. In my opinion, authors should discuss the large difference of the apparent

thickness of the domain walls when measured by PFM and SHG (Figure 1). I understand that the non-Ising character of these walls is associated to an increase of the wall thickness, but, to which extent? Is it possible to obtain an estimate?

I cannot find any obvious flaw in the analysis of the SHG data that allows an understanding of the origin of the SHG signals at ferroelectric domain walls, already reported in previous studies. But I have some trouble understanding the polar plots of the normalized SHG intensity. Authors claim that the SHG images of the domain walls are independent of the value of the polarizer angle φ respect to the crystal axis x and y , which I understand correspond to (100) and (010) of the tetragonal PZT crystal (Figure 2 and 3). In fact the SHG images do not change when $\varphi = 0$ or 90° . Thus the different dipoles in the xy plane are identified at different angles α of the analyzer (no matter φ , which I imagine is fixed and it corresponds to the red arrows in the 2D SHG maps). However, φ plays a role as it appears again in Figure 3 e, where a 3D representation of the SHG signal is drawn. I think authors should clarify this apparent contradiction. I imagine that in this case, for symmetry reasons, the two angles produce the same SHG signal. Please note that authors are also using two φ angles in Figure 4a (although the figure caption says that only 00 is used), to show again that the images are not depending on φ .

Figure 1 show SHG images of integrated intensities over α values (it is explained in the supplementary information, but I think it will help readers if this is indicated in the figure caption). Figure 2 only shows the simulations for horizontal and vertical walls, although oblique domain walls are also discussed later. Why do not show the simulations for these walls? I think it will help the reader to understand the following discussion.

I have another question regarding Figure 3. Should the reader compare b, d and f? Authors mention in the text that there is no exact correspondence ("an angular deviation of about 15° is observed between the experiments and the simulations in the case of oblique walls."), but I found that all of them are very different in intensity and I do not understand the reason of these differences. Is it because of the problems to simulate the experimental SHG intensity? I think it will be helpful for readers if this is clearer in the text.

Despite these details that authors should consider revising, the conclusions drawn seem well based on the analysis of the experimental data, and if some other authors find that there is a flaw in the arguments used, they will be able to discuss it as details are given in the manuscript and supplementary information. Therefore, I am happy to recommend the publication of this manuscript in Nature communications.

Response to the referees' comments

We would like to thank the three Reviewers for their detailed evaluation of our manuscript and their valuable comments. We are pleased about the positive opinion they have expressed on this work in view of the current ongoing research on topological structures and ferroelectric domain walls. The Reviewers put forward constructive remarks and questions. We provide in the following a point-by-point reply to their questions and the corresponding list of modifications introduced in the revised manuscript.

Comments and questions raised by referee #1

Q 1.1. SHG method is used to determine the symmetry of domain walls, but it is not able to map the detailed polar vectors within domain walls. Therefore, it is difficult to determine the type of domain walls (i.e., Néel, Bloch, etc.) by the SHG method alone. The authors determined the types of domain walls by modeling the SHG signals. The question is that: Is there a one-to-one correspondence between the domain-wall types and SHG signals in modeling? Is it possible that two or more domain-wall types (may not be Néel or Bloch) can generate a similar or the same SHG signal?

R 1.1. We agree with the referee; SHG microscopy does not provide the spatial resolution that would allow to image the precise rotation of the polar vector at the atomic scale. However, the method yields accurate information on the local symmetry and on the ferroic domain structure. Therefore SHG allows to identify domain wall types to the extent that they possess different symmetry properties. In the case of, *e.g.*, tetragonal PZT thin films with c-domains, one does not expect any SHG signal (see Supplementary figure S9). The fact that we do observe an SHG signal reveals the existence of an in-plane component of the polarization. We can therefore rule out that the domain wall is purely of Ising type. More precisely, in absence of morphotropic phase boundary regions (which are excluded in tetragonal thin films), we can conclude that there is a non-vanishing component of the polarization in the walls, directed orthogonal to the polarization axis in the parent phase, i.e., a deviation from the ideal Ising configuration.

Obviously, SHG observations cannot resolve the continuous tanh-type transition of Néel or Bloch-type walls. However, by means of local polarimetry analysis, it can precisely reveal the orientation of the overall polarization within the walls. In the case of Néel type walls, the polarization rotates across the wall, which implies the existence of an overall in-plane polarization component perpendicular to the walls, whereas a Bloch-type rotation would lead to a planar polarization along the walls. To discriminate between these two fundamental domain wall types we have modeled the SHG polarimetry signal arising from the domain boundary regions by considering a uniform in-plane polarization oriented either perpendicular to the wall (Néel-type) or parallel to it (Bloch-type). We show a one-to-one correspondence between the simulated polar plots according and the measurements conducted in $\text{Pb}(\text{Zr},\text{Ti})\text{O}_3$ thin films and LiTaO_3 bulk crystals.

Based on our simulations and their comparison with the SHG polarimetry measurements, one can therefore draw solid conclusions concerning the existence of either Néel-type or Bloch-type components, since these are clearly different from one another. However, we cannot distinguish

between the pure forms of Néel and Bloch domain walls and their mixed variants, such as a mixed Ising-Néel or mixed Ising-Bloch configuration or bichiral mixed walls. In other terms, with our experiment we can precisely determine the existence of fundamental non-Ising components and identify their type (i.e., dominating Bloch or Néel-type). However, we cannot rule out the possibility of more complex mixed domain wall types which contain an Ising component.

We thank the reviewer for pointing out that there could be complex domain wall structures which cannot be determined and that mixed domain wall types could lead to the same response. We added a paragraph in the main text and in the Supplementary Materials to discuss this point (see following text in blue color).

Implemented text in the (Text body, page 12): "The observation of Néel and Bloch-type configurations does not imply the absence of an Ising component. In uniaxial ferroelectrics, the Ising character should dominate due to electrostrictive and electrostatic effects. Therefore, a perfect Néel or Bloch like rotation of the polarization vector is unlikely. Instead, a coexistence of Ising-Néel or Ising-Bloch configurations is expected [Lee et al. Phys Rev. B 80 (2009) 060102(R)]. In this case, the amplitude of the polarization decreases during the rotation and it vanishes at the center of the wall. The possibility of opposite rotation of the polarization at each side of the wall has also been predicted. [Yudin et al. Phys. Rev. B 86 (2012) 134102] We refer to this type of mixed walls as bichiral DWs, following the terminology used by Houchmandzadeh *et al.* [J. Phys.: Condens. Matter 3, 5163 (1991)]. Our method cannot distinguish between pure forms of Néel (Bloch) DWs and mixed variants thereof, such as mixed Ising-Néel or Ising-Bloch configurations (see Supplementary Information Section III) or bichiral mixed walls. Even though SHG microscopy does not provide the spatial resolution required to image the precise rotation of the polar vector at the atomic scale, this study shows that dominant Bloch or Néel-type configurations can be revealed and discriminated by means of local SHG polarimetry."

Text provided as Supplementary Information (Section III): "III) Mixed Néel-Ising and Bloch-Ising character: Figure S13 shows 2D SHG simulations of mixed DWs in tetragonal PZT. Dark regions are expected at the center of the walls, while the overall polarization dependence of the SHG remains analogous to the variation obtained in case of pure Bloch or Néel-type walls. The dark regions are related to the centrosymmetric Ising center with zero polarization. These dark regions are not visible in our experiments because of the limited lateral resolution."

Figure S1: DWs' SHG simulated for a rectangular c-domain in tetragonal PZT. The mixed character of the walls has been considered in the 2D simulations.

Q 1.2. Based on Figure 1 and 5, it appears the thickness of domain walls is in the range of 100~1000 nm. This value is very large compared to those determined from TEM observation which show that the thickness of domain walls is generally around several nanometers [e.g., Jia CL et al, Nat Mater, 7, 57-61 (2008)]. Is it due to the low resolution of SHG?

R 1.2. Determining precisely the thickness of ferroelectric domain walls is a very challenging task. In the literature one can find reports on several experimental methods that have been employed, in vain, to obtain this information. This is summarized, e.g., in the book by A.K. Tagantsev et al. "Domains in ferroelectric crystals and thin films", Table 6.1.1 (pp 274-277), where numerous data on the width of domain walls in different ferroic systems are reviewed. Variations from a few atomic cells up to several microns have been reported. However, it has become increasingly clear nowadays that high-resolution TEM measurements give the most accurate estimate of the wall size (typically a few atomic cells), while optical methods will always show the walls as broadened regions of micron size.

In our confocal nonlinear optical microscopy experiment, the convolution of a virtually zero DW size (a few unit cells to a few nanometers, as mentioned by the Reviewer) with the width of the focused fundamental laser beam (FWHM = 1000 nm) makes the DWs appear as broadened regions replicating the size of the fundamental beam. The polarimetry analysis allowing for the local symmetry analysis was made possible by exploiting this broadening effect. Nevertheless, in the Supplementary Figure S5 we demonstrate that an analysis of the SHG profile at domain walls is not relevant for drawing information on the domain wall size. Such an analysis could, however, be useful as a resolution test for the instrument as explained by Denev *et al.* [J. Am. Ceram. Soc., 94 (2011) 2699].

Q 1.3. For PZT thin films, the authors designed two artificial domain shapes by using a PFM tip, where non-ising domain walls were observed. However, these two domain shapes cannot be achieved by naturally. Are the types of domain walls highly related to the geometric shapes of the domains? Is it possible to observe a non-ising domain wall in naturally formed domain patterns in PZT? To answer this question, the authors need to do SHG experiments on a PZT thin film without written domains.

R 1.3. We agree that such a comparative study of as-grown domains vs artificially patterned domain structures would help in better understanding the microscopic origin of non-Ising domain walls. Unfortunately it is practically impossible to perform such a comparison given the difficulty to obtain both natural and artificial 180° domain walls in PZT. This is due to the fact that as-grown tetragonal PZT films generally exhibit a uniform single domain structure (the whole film is a c^- -domain when grown on SrRuO₃ electrode, or a uniform c^+ -domain when grown on LaSrMnO₃ buffer layer. See Methods). In general, it is hard to control the intrinsic domain state in a way that would allow both, the presence of a bottom electrode (necessary for voltage application), and multidomain states (*i.e.*, no bottom electrode). In addition, the intrinsic multidomain states in thin films generally result in small (about 100 nm), densely packed domains, from which it would be difficult to resolve SHG at the domain walls. The purpose of our artificially prepared domains was to obtain well-defined walls, spaced out far enough so that there is an unambiguous SHG signal from any given, individual wall.

Concerning the correlation between the domain shapes and the wall types, we have addressed this question by investigating the orientation of the (artificially written) walls with respect to the crystal axes (straight vs oblique walls). We demonstrate that in the case of tetragonal PZT as well as in Y-oriented walls in trigonal lithium tantalate, the effect is insignificant. Yet, based on theoretical predictions (see,

e.g., studies conducted by Morozovska *et al.*) we expect strong variations to occur in systems exhibiting significant variations of the flexoelectric coupling coefficients (sign and value), in metastable walls such as X-oriented walls in lithium tantalate, or in cylindrical domains. The study of such complex systems is beyond the goal of this manuscript which focuses mainly on revealing deviations from the ideal Ising configuration of model 180° walls in uniaxial ferroelectrics. We hope, however, that our study will stimulate further investigations on more exotic systems.

Q 1.4. The origin of non-ising domain walls is not clearly discussed in the present paper. The authors need to give a specific discussion on the origin of non-ising domain walls for each cases (i.e., PZT and LT). The Ising-like domain walls should also exist in PZT and LT for certain conditions. So, which conditions favor Ising-domain walls and which favor Neel or Bloch domain walls? This issue is important for the design of ferroelectric devices based on domain walls.

R 1.4. The existence of mixed domain walls (Bloch-Ising and Néel-Ising) is now discussed in the manuscript as clarified in our response R1.1.

We agree that the origin of non-Ising domain walls was not discussed in the previous version of the manuscript. We hope that the paragraph we implemented in the Discussion section satisfies the reviewer. However, we would like to mention that our study is primarily about the experimental observation of chiral and non-Ising walls. The provided discussion is based purely on previously reported theoretical studies. Furthermore, only a small fraction of these predictions of the non-Ising character at ferroelectric domain walls explains the origin of the effect in selected materials.

Implemented paragraph in the Discussion section (page 14 of the revised manuscript): "Little is known so far about the origin of the non-Ising structure of DWs. This deviation could be explained by localized electro-mechanical coupling. For example, lattice distortions caused by oxygen octahedron rotations in perovskite oxides are capable of exhibiting improper ferroelectricity (*e.g.*, at elastic twins) due to the coupling of the flexoelectric effect (polarization induced by strain gradient) and rotostriction. [Morozovska *et al.* Phys Rev B 85 (2012) 094107] However, this effect should play a minor role in uniaxial ferroelectrics like those investigated in this study. Only two theoretical studies that we know of have investigated the origin of the deviation of nominally uncharged 180° DWs from the Ising-type model.[Yudin *et al.* Phys. Rev. B 86 (2012) 134102; Gu *et al.* Phys. Rev. B 86 (2014) 174111] These studies have shown that the domain wall type in tetragonal BaTiO₃ is determined by the competition between the depolarization and the flexoelectric fields."

Comments and questions raised by referee #2:

Q 2.1. The authors consider only Ps chirality and do not think over the oxygen octahedron rotations (OORs), which have been pointed out to play the important role at DWs in some perovskite ferroics, as discussed in the references below. The OOR could cause the symmetry lowering at DWs. The authors should examine the effect and make comments on the possibility of this kind of symmetry lowering when they analyze the SHG anisotropy.

1-1. Mark Calleja et al. *J. Phys.: Condens. Matter* **15** (2003) 2301.

1-2. Anna N. Morozovska et al. *PHYSICAL REVIEW B* **85**, 094107 (2012).

1-3. S. Van Aert, et al. *Adv. Materials* **24**, 523 (2012).

R 2.1. The Referee is perfectly right. Lattice distortions such as oxygen octahedron rotations (OOR) are capable of breaking the symmetry, leading to a non-vanishing polarization at domain boundaries (as described in the references kindly provided by the Reviewer). We thank the Referee for bringing this point to our attention. We have followed his/her advice, and we have implemented a Discussion section in which the possible origins of polar domain walls are discussed (see manuscript, page 14). The discussion includes the flexoelectric coupling and rotostriction that are the driving mechanisms through which OOR leads to a spontaneous polarization at domain walls. However, we would like to point out that the effect of OOR is dominant mostly at surfaces and interfaces (antiphase boundaries, twins, multilayers), as well as in incipient ferroelectrics such as CaTiO₃ and SrTiO₃. Uniaxial ferroelectrics like those investigated in our study, in particular bulk crystals and strongly tetragonal films, should not exhibit a significant OOR.

Q 2.2. In this research, the reflection-type SHGM is used. However it is well know that the analysis of this type of experiments is very complicated, because various factors affect the results (see the references below). For example, are SH waves generated when the fundamental waves (FWs) go forward in the sample, or they go back after being reflected at the substrate (PZT) or the bottom surface (LT)? The former is the inverse coupling case with the wave number miss-fit, $\Delta k_i = k(2\omega) + 2k(\omega)$, and the latter is the regular coupling case, $\Delta k_r = k(2\omega) - 2k(\omega)$. The two cases give different behavior of SH waves. The authors should discuss this problem in the manuscript.

2-1. E. Sipe et al. *PRB*, **35**, 9091 (1987).

2-2. S. W. Liu et al. *JAP*, **101**, 104118 (2007).

R 2.2. We agree with the Referee in that both phase mismatch [Liu *et al.*] and surface contributions [Sipe *et al.*] affect the intensity of the SHG signal in the reflection measurement geometry.

Nonetheless, we would like to stress that the main results presented in this study are based on the polarization features of the SHG signal (*i.e.* the construction/interpretation of the susceptibility tensor), not on its amplitude. The former (*i.e.* the shape of the SHG polar plots) is not affected by phase-mismatch effects, due to the specific crystal configurations we chose for the experiments. For instance, for LiTaO₃ we chose *z* as the propagation direction where both the SH and the fundamental wave fields are polarized in the *xy* plane. Given that the crystal is uniaxial and that *z* is its optic axis, in this configuration the refractive indices of the (fundamental wave and SHG) fields do not depend on their polarization in

the xy plane. Accordingly, the wavevector mismatches for all possible SHG configurations (both forward and backward) in this geometry are independent of the field polarization angles (φ , α). As a result, the phase-mismatch and the presence of multiple SHG processes do not affect the shape of the SHG polar plots, although they have indeed an impact on the amplitude scales of the SHG plots. We realized this was an important point not clearly mentioned in the original manuscript, which we now explicitly deal with in the revised text (see more details below).

On the other hand, surface SHG can indeed lead to new tensor components at surfaces and interfaces – as compared to the bulk – resulting in a deformation of the SHG polar plots. In the specific case of LT this effect can be essentially removed in the experiments by focusing the fundamental beam deep into the bulk (100 μm away from the surface). Such considerations do not apply to the experiments on thin-film PZT crystals, where the thickness of the film is less than the Rayleigh range of the pump beam ($\sim 3 \mu\text{m}$). Nevertheless, given that the surface SHG signal is homogeneous in the xy-plane of the film, the signal collected at domains and domain walls should be equally affected by the surface contribution. Therefore, the surface contribution can be viewed as a background signal that can be eliminated by following the procedure described in the Supplementary Figure S.3. in the case of PZT films.

We believe that the question raised by the referee is important as it points out possible experimental difficulties. Accordingly, we have implemented the following section in which we discuss these issues, following his/her recommendation: “*Supplementary paragraph, Section I. (6). On the SHG detection in the reflection geometry: In this study, the nondestructive reflection-type SHG geometry is employed. It is worth noting that the interpretation of the axial SHG signal requires specific precautions regarding possible phase matching effects affecting the SHG intensity as well as additional SHG contributions arising from symmetry breaking at the crystal surfaces.*

Phase mismatch. In reflection geometry, the back-reflected forward SHG process is characterized by a wavevector mismatch $\Delta k_r = k(2\omega) - 2k(\omega)$, where $k(\omega)$ and $k(2\omega)$ are the wavenumbers of second-harmonic and fundamental waves, respectively. Both are ordinarily polarized in our case. The very same configuration can also support backward SHG with a wavevector mismatch $\Delta k_i = k(2\omega) + 2k(\omega)$, where $k(2\omega)$ and $k(\omega)$ are the same quantities defined above. In principle, both forward and backward signals contribute to the overall SHG intensity, although the former might be expected to be the dominant effects, in light of its larger coherence length (1.55 μm as opposed to a value of $\sim 50 \text{ nm}$ for backward SHG). Therefore, the phase mismatch affects the total SHG efficiency. However, when the electromagnetic wave propagates along the z-axis, both the second-harmonic and the fundamental waves are ordinary polarized (*i.e.*, polarized in the xy-plane). In this case, the values of Δk_r , Δk_i , and of the propagation distance (given by the crystal thickness) are constants. Their contribution affects the SHG intensity but it should not alter the shape of the SHG polar plots on which our conclusions are based. Note that this would not be the case if the experiments were done with waves propagating along x or y.

Surface SHG. Early studies have shown that SHG could originate at surfaces, even in centrosymmetric materials. This is due to the strong discontinuity of the electric field induced by the symmetry breaking at the surface. The specific nonlinear susceptibility tensors associated to surfaces may lead to a strongly different SHG response compared to that expected in the bulk of a given material. In particular, a specific nonlinearity (polar plot) and an increased intensity are expected at surfaces. These new tensor components at the surface – as compared to the bulk – may result in a deformation of the SHG polar plots. For this reason, in the article we discuss only the results obtained in the case when the fundamental laser beam is focused into the sample

volume in the specific case of lithium tantalate. Moreover, since the surface SHG should be generated equally in the xy-plane, the surface contribution at the domain wall regions should be identical to that arising from the domains. Therefore, the specific SHG anisotropy at the domain wall regions (as compared to the domains) is unrelated to the uniform surface SHG. This background signal is subtracted in the case of PZT films, by following the method described in the supplementary figure S.3. This contribution is insignificant in the case of LiTaO₃ bulk crystals, where the focus point is at 100 μm below the surface and the background signal is negligible.”

Q 2.3. It has been also pointed out that the internal observation of the sample using the SHGM depends strongly upon the sign of Δk . (see the reference below). In the case of Δk positive, SHG intensities from the inside are very weak, which makes the observation difficult. In LT crystals, all SHG tensor components are known to be positive and a special care is necessary for the observations. The authors should make some comments on it.

3-1. J. Kaneshiro et al. J. Appl. Phys. 104, 054112 (2008).

R 2.3. We agree with the Referee. If Δk (as defined by eq. (2) in Kaneshiro et al. J. Appl. Phys. 104, 054112 (2008)) has a positive sign, the expected SHG signal is small. Our measurements confirm that in the case of lithium tantalate, and given our measurement geometry, Δk is positive. The measured SHG intensity trend as a function of depth in our experiments (*cf.* supplementary figure below) is in good agreement with typical SHG results as reported, e.g., by Kaneshiro et al. J. Appl. Phys. 104, 054112 (2008) concerning both, the SHG enhancement at the surface, and the comparatively ultra-low SHG signal inside the crystal. In spite of the low intensity, $\Delta k > 0$ favors a high signal-to-background ratio at the domain walls which allows for the detection of local nonlinearities and the analysis of their polarization.

We follow the Referee’s advice and we add the following discussion and a supplementary figure: Supp. Info. I. (5) “A three-dimensional (3D) investigation of the DWs is conducted by means of 3D SHG microscopy [Uesu *et al.* *Ferroelectrics* 304 (2004) 99; Kaneshiro *et al.* *J. Appl. Phys.* 104, 054112 (2008)] in a nearly stoichiometric periodically poled lithium tantalate bulk crystals in which periodic domains have been periodically poled (alternated c^+/c^- strip domains) by electric field throughout the 500mm sample thickness. Due to the elongated shape of the laser beam in the z-direction (about 3 μm, as opposed to 1 μm along (xy)), the focal position is changed by moving the sample along the axial direction z in steps of about 10 μm. Due to the limited thickness of the PZT film (< 3mm), only LT crystals could be probed by this technique.

Figure S6 (a,b) shows the polar plots recorded at different depths in the c-domains (background signal) and in the domain wall regions. In spite of a strong decrease of the absolute SHG intensity in the interior of the sample with respect to the surface due to a positive value of the SHG mismatch Dk_r [Kaneshiro et al. *J. Appl. Phys.* 104, 054112 (2008)], the local nonlinearity of the walls is clearly visible. We exploit the good signal-to-background ratio at the walls (SHG at c-domains is negligible) to construct 3D images of the wall regions as shown in Figure S6 (c).

Figure S 6: (a,b) Depth-dependent SHG polar plots and (c) 3D SHG Profile at 180° DWs in LiTaO₃.

Q 2.4. What happens at the boundaries of the x-axis elongated DW and the y-axis elongated one? (Fig. 1(e) for example).

R 2.4. In our experiment, it is unfortunately not possible to draw unambiguous conclusions concerning the intersection region between two walls, given the difficulty to resolve the SHG of this small region among the signal arising from the adjacent walls. Therefore, the measured SHG at the boundary of two, e.g. orthogonal walls in PZT, results essentially as the superposition of the SHG of the two walls. As discussed in the response R1.2 addressed to Reviewer 1, the SHG spatial profile at the domain walls is essentially determined by the width of the fundamental beam, *i.e.* $\sim 1 \mu\text{m}$ in our experiment. Thus, if the boundary region is smaller than the SHG profile, it is impossible to resolve its details with this technique. To our knowledge, this exciting question concerning the intersection domain walls was addressed so far only in a theoretical study by Scrymgeour et al. [Phys. Rev. B 71 (2005) 184110] in which the authors analyze the internal structure of Y and X-oriented walls in lithium niobate and lithium tantalate. There, it was shown that the intersection of the wall segments forming a hexagonal domain exhibits a different internal structure compared to the adjacent walls.

Q 2.5. What factors determine the type of DWs, Ising type, Neel type, or Bloch type?

R 2.5. Concerning the factors that may affect the type of domain walls, very little is known at the moment. While the existence of Néel and Bloch-type wall was reported in quite a few theoretical studies, only two papers that we know of account for the origin of the deviation from the Ising-type [Yudin *et al.* Phys. Rev. B 86 (2012) 134102; Gu *et al.* Phys. Rev. B 86 (2014) 174111]. Among the possible effects that may determine the wall type, the competing depolarization field and the flexoelectric effect. (see revised manuscript, Discussion, page 13). This thrilling question is unfortunately beyond the scope of our experimental study which reports on the observation of non-Ising and chiral domain walls.

Q 2.6. For observations of DWs in LT, the domain structure was patterned using the metal electrodes, and then the electrodes were removed by the chemical etching technique. In this case, it is expected that DWs are topologically different from the up and down domains. Does this affect the SHG intensities?

R 2.6. We agree with the Reviewer also on this point. A soft selective etching of the domains may occur during the chemical removal of the electrodes. This slight corrugation of the surface may affect the intensities in interferometric SHG through a step effect. We would nevertheless like to stress that our results are obtained in the non-interference SHG mode. Furthermore, in the specific case of LT, the fundamental wave is focused well inside the sample (in order to suppress surface effects as discussed in R2.2). Therefore, slight difference in surface topology at c^+ and c^- domains do not affect the SHG intensities in our measurements as illustrated, e.g., in Figure 4 (a) and (b)) as well as the supplementary Figure provided in our response R2.3.

Q 2.7. In Fig.1, the directions of P and A are not so clear. It should be noted in the caption.

Q 2.8. In the caption of Fig.1, "lead titanate" should be "PZT".

R 2.7 & 2.8 We are grateful to the Reviewer for the careful reading of our manuscript. We have updated the figure caption (see below) according to his/her recommendation:

"Figure 2: Detection of localized second-harmonic emission at DWs in lead zirconate titanate thin films. (a) Schematic representation of the SHG experimental setup. Ferroelectric domains (c^+ and c^- domains) of different geometric shapes are written in the film by applying a bias voltage through a conductive PFM tip. The domain structure imaged by means of PFM is displayed for film regions with **(b)** right triangle and **(c)** rectangle-shaped domains. The corresponding nonlinear optical images reveal a localized optical signal at the walls surrounding the **(d)** triangular and **(e)** rectangular domains. The bars in all images correspond to a scale of $2\ \mu\text{m}$. **(f)** The fundamental wavelength is 800 nm while the localized emission **(g)** exhibits a wavelength of 400 nm, corresponding to the half of the fundamental wavelength. As shown in panel **(h)**, the emission intensity also displays a quadratic dependence on the power of the fundamental wave (FW). The SHG images represent the data recorded at a polarizer angle $\varphi = 0^\circ$ integrated over the analyzer angles α ."

Comments and questions raised by referee #3:

Q 3.1 Authors only analyze two cases. First, Lead Zirconate Titanate (note that the word “zirconate” is missing in some parts of the text: abstract, figure 1 caption...) thin films. And, secondly, Lithium Tantalate single crystals. The choice of these specific materials is not clearly stated in the text, but it seems that the PZT film is only used to show the experimental demonstration of the non-Ising nature of ferroelectric domain walls. In this case authors fail to cite any previous calculations on domain walls that predict their Néel-type character in tetragonal PZT. Probably these references do not exist, but then, why choose tetragonal PZT if not previous simulations were reported? Being a thin film, it is a pity that we do not have the opportunity to see the 3D study of the domain wall, similar to the one performed for LT single crystal. In this second case we can compare the experimental results with theoretical predictions. In my opinion, authors should discuss the choice made and the applicability of this method to the study of the nature of ferroelectric walls in other compositions and materials in order to strengthen the conclusions drawn.

R 3.1 We have corrected the denomination of PZT where the word zirconate was missing, *i.e.*, in the caption of figure 1 (see *R 2.7 & 2.8*) as well as in the abstract: "Here we probe the internal structure of 180° ferroelectric DWs in lead zirconate titanate (PZT) thin films and lithium tantalate (LT) bulk crystals...".

We perfectly agree with the remark of the Reviewer, pointing out that the choice of the studied systems was not sufficiently motivated in the manuscript. In particular, two important references to theoretical studies predicting the Néel character of domain walls in PZT 20:80 were missing. We thank the reviewer for pointing out this omission. We have updated the manuscript accordingly by inserting the following paragraph (page 4): "To prove the general applicability of our approach to identify a possible deviation of ferroelectric DWs from the ideal Ising-type configuration, we focus on nominally uncharged 180° domain DWs in two fundamentally different crystals: the one is a tetragonal lead zirconate titanate thin film, the other a lithium tantalite bulk trigonal crystal. This choice is motivated by theoretical studies predicting the existence of non-Ising walls with an internal structure specific to each system. A Bloch-type configuration has been predicted for Y-oriented walls in crystals of the lithium niobate family [Scrymgeour *et al.*, Phys. Rev. B 71 (2005) 184110], while Morozovska [Ferroelectrics, 438 (2012) 3] and Eliseev *et al.* [Eliseev *et al.*, Phys. Rev. B 85 (2012) 045312] show that Néel-type DWs are expected in \$\text{PbZr}_{0.2}\text{Ti}_{0.8}\text{O}_3\$. Moreover, \$\text{PbZr}_{0.2}\text{Ti}_{0.8}\text{O}_3\$ is often used as a stand-in system for pure \$\text{PbTiO}_3\$ in which Néel-type DWs are expected [Lee *et al.* Phys Rev. B 80 (2009) 060102(R); Behera *et al.* J. Phys.: Condens. Matter 23 (2011) 175902]."

The method reported in this study is in principle applicable to analyze DWs in both thin films and bulk materials, with the additional capability of three-dimensional SHG mapping of the wall profile in the latter case. Yet, we share the frustration of the Referee about not being able to probe the 3D configuration in thin films because of the limited resolution in the depth direction ($\approx 3 \mu\text{m}$).

Q 3.2 Another interesting factor is the fact that none of the measurements at a local scale as Transmission Electron Microscopy or Piezoresponse Force Microscopy has been able to unveil the character of these walls. In my opinion, authors should discuss the large difference of the apparent thickness of the domain walls when measured by PFM and SHG (Figure 1). I understand that the non-

Ising character of these walls is associated to an increase of the wall thickness, but, to which extent? Is it possible to obtain an estimate?

R 3.2. Both Reviewers 1 and 3 brought up the important question of the thickness of the domain walls. As explained in R1.2., the SHG technique images the convolution of a virtually zero DW size (Dirac peak) with the width of the focused fundamental laser beam (approximated by a Gaussian function with a FWHM ≈ 1000 nm). Therefore the DWs appear as broadened regions replicating the size of the fundamental beam. In general, the spatial resolution of SHG is not sufficient for a detailed and complete analysis of the wall profile, especially as far as the domain wall width is concerned. Yet, the apparent broadening effect can be used to extract important information on the domain wall structure, because it allows for the precise analysis of the local symmetry and displays the local ferroelectric order in a region smaller than the lateral resolution.

Q 3.3 I cannot find any obvious flaw in the analysis of the SHG data that allows an understanding of the origin of the SHG signals at ferroelectric domain walls, already reported in previous studies. But I have some trouble understanding the polar plots of the normalized SHG intensity. Authors claim that the SHG images of the domain walls are independent of the value of the polarizer angle ϕ respect to the crystal axis x and y , which I understand correspond to (100) and (010) of the tetragonal PZT crystal (Figure 2 and 3). In fact the SHG images do not change when $\phi = 0$ or 90° . Thus the different dipoles in the xy plane are identified at different angles α of the analyzer (no matter ϕ , which I imagine is fixed and it corresponds to the red arrows in the 2D SHG maps). However, ϕ plays a role as it appears again in Figure 3 e, where a 3D representation of the SHG signal is drawn. I think authors should clarify this apparent contradiction. I imagine that in this case, for symmetry reasons, the two angles produce the same SHG signal. Please note that authors are also using two ϕ angles in Figure 4a (although the figure caption says that only 00 is used), to show again that the images are not depending on ϕ .

R 3.3 We agree with the Reviewer that the polarization of the fundamental wave affects the second-harmonic response. We understand that the way the invariance with ϕ was presented in the manuscript may have appeared to be a general rule, while it in fact only applies to the two angles: $\phi = 0^\circ$ and $\phi = 90^\circ$, and only in the case of horizontal and vertical DWs for the specific symmetry of the considered materials.

As shown in the supplementary figure that we append below, when the polarizer angles are fixed at $\phi = 0^\circ$ and $\phi = 90^\circ$, only the SHG intensity varies and not the anisotropy of the signal. Both, the experiments as well as the analytical derivation of the SHG intensity (calculated using the Supplementary equations S12 and S15 in the case of PZT) show that the angle of the fundamental wave has an effect on the SHG intensity at 0° and 90° , but not on its anisotropy. Therefore, in Figure 4 we have represented the SHG microscopy results at different polarizer and analyzer angle settings but the polar plots are shown only in the case of $\phi = 0^\circ$, since the case $\phi = 90^\circ$ leads to the same polar shape.

Fig.1: (a) Simulated and (b) measured polar plots at horizontal and vertical domain walls in tetragonal lead zirconate titanate. Both the experiment and the simulations (based on the analytic form of the SHG expected at Néel and Bloch walls) show a uniaxial anisotropy of the SHG polarization, with a maximum intensity perpendicular to the walls in the case of Néel-type configuration. When the polarization of the fundamental wave is turned from $\varphi = 0^\circ$ to $\varphi = 90^\circ$ the SHG intensity varies but the anisotropy is preserved.

We follow the advice of the Reviewer and add the following clarification in the text (see revised manuscript, page 7): "(...) These results indicate a preferential SHG polarization perpendicular to the walls in the case of Néel-type configuration and parallel to them in the case of Bloch-type DWs. When the polarization of the fundamental beam is turned from $\varphi = 0^\circ$ to $\varphi = 90^\circ$, only the SHG intensity varies at horizontal and vertical DWs and not the anisotropy of the signal."

The reviewer's comments made us realize that the transition from the 3D representation (spherical coordinates) of the simulated SHG to the 2D polar plots (of simulations and experiments) was not sufficiently clear. By choosing a different representation, it becomes easier to convey the complex dependence of the SHG signal on the polarizer angles. Therefore we are now including a more detailed explanation and a more intuitively understandable visual presentation of the 3D data in the revised manuscript.

In the revised text, we present the simulated SHG polarimetry in a cylindrical coordinate system ($I^{\text{SHG}}, \alpha, z = \varphi$) instead of using spherical coordinates. By using this modified representation, horizontal plane cuts (corresponding to planes at $z = \text{const.}$) yield the 2D polar plots at a given φ angle. The following corrections have been implemented in the text (page 9) "The calculated SHG intensity is presented in the cylindrical coordinate system ($I^{\text{SHG}}, \alpha, z = \varphi$). Different visualization angles are provided in Supplementary Movie 1. This 3D visualization allows for the full representation of the SHG polarimetry from which polar plots can be extracted through plane cuts at constant $z = \varphi$. The experimental polar plots measured at $\varphi = 0^\circ$ (scattered dots in Fig. 3b,d) can thus be related to the simulated plots displayed in Figure 3f. Note that due to the experimental difficulty to account for absolute SHG intensities, the normalized SHG intensities are displayed. By doing so, the SHG anisotropy can be identified immediately. As a result, any comparison between the experimental and the simulated data shown here should only refer to the shape and orientation of the polar plots, and not the absolute SHG intensity."

We have revised figures 3 and 4 (see updated figures below) and movies 1 and 2 accordingly.

Figure 3: Polarimetry analysis of the DWs' SHG signal in tetragonal PZT. Experimental SHG microscopy results for different polarizer and analyzer settings in c-domains with (a) rectangular and (c) triangular shapes. The white bars correspond to a scale of $2\ \mu\text{m}$. A systematic local SHG polarimetry analysis is conducted at horizontal, vertical, and oblique walls. The corresponding polar plots (scattered dots) of the normalized SHG intensity measured at a fundamental polarization angle $\varphi = 0^\circ$ are shown in case of (b) rectangles and (d) angle triangles. The continuous lines are fits of the experimental data to the analytic expression of the SHG intensity expected for Néel-type walls with horizontal, vertical, or oblique orientations. (e) 3D SHG polarimetry simulated in the case of a planar ferroelectric polarization along the y-axis (Néel-type), represented in cylindrical coordinates ($x = I^{\text{SHG}} \cos\alpha$, $y = I^{\text{SHG}} \sin\alpha$, and $z = \varphi$). The z-axis has been scaled with a factor 10 for better visibility. The color map represents the SHG intensity in arbitrary units and the continuous lines result from plane cuts at $\varphi = 0^\circ$ (red color), $\varphi = 90^\circ$ and 180° (gray). The normalized intensity of the calculated polar plots at $\varphi = 0^\circ$ is displayed in panel (f) to facilitate the comparison with the experimental results.

Figure 4: SHG Polarimetry analysis in LiTaO₃. (a) SHG image of typical hexagonal c-domains in nearly stoichiometric LT measured at polarizer angles $\alpha = 0^\circ$ and $\phi = 0^\circ$. (b) The polar plot measured at the HDWs of the hexagon shows a maximum SHG polarization along the walls. The bars in all images correspond to a scale of 5 μm . (c) The 3D SHG polarimetry has been calculated assuming the existence of a planar ferroelectric polarization oriented either parallel (Bloch) or perpendicular to the walls (Néel). The simulation results are displayed both for horizontal (\parallel y-axis) and oblique walls ($\pm 60^\circ$ with respect to y-axis). The color map denotes the normalized SHG intensity in arbitrary units, and the continuous lines represent the SHG polar plots at $\phi = 0^\circ$ (red color) $\phi = 90^\circ$ and 180° (gray).

Q 3.4 Figure 1 show SHG images of integrated intensities over α values (it is explained in the supplementary information, but I think it will help readers if this is indicated in the figure caption).

R 3.4 We have updated the figure caption according to the recommendation of the Referee:

Figure 5: Detection of localized second-harmonic emission at DWs in lead zirconate titanate thin films. (a) Schematic representation of the SHG experimental setup. Ferroelectric domains (c^+ and c^- domains) of different geometric shapes are written in the film by applying a bias voltage through a conductive PFM tip. The domain structure imaged by means of PFM is displayed for film regions with (b) right triangle and (c) rectangle-shaped domains. The corresponding nonlinear optical images reveal a localized optical signal at the walls surrounding the (d) triangular and (e) rectangular domains. The bars in all images correspond to a scale of 2 μm . (f) The fundamental wavelength is 800 nm while the localized emission (g) exhibits a wavelength of 400 nm, corresponding to the half of the fundamental wavelength. As shown in panel (h), the emission intensity also displays a quadratic dependence on the power of the fundamental wave (FW). The SHG images represent the data recorded at a polarizer angle $\phi = 0^\circ$ integrated over the analyzer angles α .

Q 3.5 Figure 2 only shows the simulations for horizontal and vertical walls, although oblique domain walls are also discussed later. Why do not show the simulations for these walls? I think it will help the reader to understand the following discussion.

R 3.5 We have followed the advice of the Reviewer and have included the 2D simulations of oblique walls. Nevertheless, we believe that the comparison of polar plots in a wall with different orientations give a more comprehensive information than the SHG images. The simulation results are provided for the convenience of the Reader as a supplementary figure S10:

Figure S 10: 2D simulations of the SHG emission at the boundary regions between c-domains in tetragonal PZT. The simulations are performed for a right angle triangle, assuming DWs with Néel-type Bloch-type internal structure. The variation of the SHG intensity at horizontal (HDW), vertical (VDW) and oblique (ODW) DWs is displayed at different polarizer and analyzer angles (ϕ and α , respectively).

Q 3.6 I have another question regarding Figure 3. Should the reader compare b, d and f? Authors mention in the text that there is no exact correspondence (“an angular deviation of about 15° is observed between the experiments and the simulations in the case of oblique walls.”), but I found that all of them are very different in intensity and I do not understand the reason of these differences. Is it because of the problems to simulate the experimental SHG intensity? I think it will be helpful for readers if this is clearer in the text.

R 3.6 Quantitative evaluations of the SHG intensity are difficult as they require relative measurements with respect to a reference system (e.g., quartz or ammonium dihydrogen phosphate) and specific measurement geometry. Nevertheless, the qualitative probe of the susceptibility tensor elements (e.g., SHG anisotropy analysis) provides valuable information on the symmetry. We confirm that panels b, d and f in Figure 3 in the previous version were presented for comparison. But only the anisotropy of the SHG signal should be examined, and not the absolute SHG intensity.

We follow the advice of the Referee and implement the following explanations in the revised manuscript to help the Reader (page 9): "Note that due to the experimental difficulty to account for absolute SHG intensities, the normalized SHG intensities are displayed. By doing so, the SHG anisotropy can be identified immediately. As a result, any comparison between the experimental and the simulated data shown here should only refer to the shape and orientation of the polar plots, and not the absolute SHG intensity."

In addition, in the revised version of Figure 3, we provide the normalized polar plots in panel f to help comparison with the normalized experimental data provided in panels b and d.

We also add in the Methods section: "The SHG intensity is calculated by taking $\epsilon_0 E_0 = 1$ and it is presented in arbitrary units".

REVIEWERS' COMMENTS:

Reviewer #1 (Remarks to the Author):

The reviewer is satisfied with the responses and revisions.

Reviewer #2 (Remarks to the Author):

The referee read closely the responses of the authors and their revised manuscript, and found that the authors replied precisely and rigorously to all his questions and comments. He thinks that the revised manuscript will be now able to be published in Nature Communications as it is.

Reviewer #3 (Remarks to the Author):

Authors have taken into account all referees' comments and make corrections in the text and figures accordingly. I think this has improved the paper as a whole. Weak points of the paper has been addressed and needed clarifications of the results have been added. The interest of this work for the community stands, and, therefore, I recommend the publication of this manuscript in Nature communications.